Urban park characteristics, genetic variation, and historical demography of white-footed mouse (Peromyscus leucopus) populations in New York City

Munshi-South Jason 1 jason@NYCevolution.org
Nagy Christopher 2
1 Department of Biological Sciences and the Louis Calder Center—Biological Field Station, Fordham University , Armonk, NY , USA
2 Mianus River Gorge Preserve , Bedford, NY , USA
Amos William
Electronic publication date: 2014 Mar 13
Publication date: 2014
Volume: 2
Electronic Location ID: e310
Received 2013 Dec 23; Accepted 2014 Feb 25
Copyright: © 2014 Munshi-South et al.
Copyright year: 2014
Copyright holder: Munshi-South et al.
License: This is an open access article distributed under the terms of the Creative Commons Attribution License, which permits unrestricted use, distribution, and reproduction in any medium, provided the original author and source are credited.
License URL: https://creativecommons.org/licenses/by/3.0/

Keywords: Population genetics, Genetic variation, Mitochondrial DNA, Urban ecology, Sex-biased dispersal, Selective sweep, Genetic bottleneck, Historical demography, Urban evolutionary biology, Peromyscus leucopus

Funding: National Science Foundation DEB 0817259 National Institute of General Medical Sciences/National Institutes of Health 1R15GM099055-01A1 This work was funded by grants from the National Science Foundation (DEB 0817259) and National Institute of General Medical Sciences/National Institutes of Health (1R15GM099055-01A1) to Jason Munshi-South. The funders had no role in study design, data collection and analysis, decision to publish, or preparation of the manuscript.

==============================
Severe fragmentation is a typical fate of native remnant habitats in cities, and urban wildlife with limited dispersal ability are predicted to lose genetic variation in isolated urban patches. However, little information exists on the characteristics of urban green spaces required to conserve genetic variation. In this study, we examine whether isolation in New York City (NYC) parks results in genetic bottlenecks in white-footed mice (Peromyscus leucopus), and test the hypotheses that park size and time since isolation are associated with genetic variability using nonlinear regression and information-theoretic model selection. White-footed mice have previously been documented to exhibit male-biased dispersal, which may create disparities in genetic variation between males and females in urban parks. We use genotypes of 18 neutral microsatellite data and four different statistical tests to assess this prediction. Given that sex-biased dispersal may create disparities between population genetic patterns inferred from bi- vs. uni-parentally inherited markers, we also sequenced a 324 bp segment of the mitochondrial D-loop for independent inferences of historical demography in urban P. leucopus. We report that isolation in urban parks does not necessarily result in genetic bottlenecks; only three out of 14 populations in NYC parks exhibited a signature of a recent bottleneck at 18 neutral microsatellite loci. Mouse populations in larger urban parks, or parks that have been isolated for shorter periods of time, also do not generally contain greater genetic variation than populations in smaller parks. These results suggest that even small networks of green spaces may be sufficient to maintain the evolutionary potential of native species with certain characteristics. We also found that isolation in urban parks results in weak to nonexistent sex-biased dispersal in a species known to exhibit male-biased dispersal in less fragmented environments. In contrast to nuclear loci, mitochondrial D-loop haplotypes exhibited a mutational pattern of demographic expansion after a recent bottleneck or selective sweep. Estimates of the timing of this expansion suggest that it occurred concurrent with urbanization of NYC over the last few dozens to hundreds of years. Given the general non-neutrality of mtDNA in many systems and evidence of selection on related coding sequences in urban P. leucopus, we argue that the P. leucopus mitochondrial genome experienced recent negative selection against haplotypes not favored in isolated urban parks. In general, rapid adaptive evolution driven by urbanization, global climate change, and other human-caused factors is underappreciated by evolutionary biologists, but many more cases will likely be documented in the near future.

Introduction

Populations in fragmented habitats are predicted to lose genetic variation due to drift and local adaptation through natural selection (Varvio, Chakraborty & Nei, 1986), although this decline may be opposed by gene flow and mutations that add new genetic variants to individual populations (Slatkin, 1987). If sufficiently severe, fragmentation promotes a cycle of reduced population size, inbreeding, and loss of genetic variation (Ellstrand & Elam, 1993). The relative importance of genetic variation in this ‘extinction vortex’ has been widely debated (Ashley et al., 2003), but the magnitude of inbreeding depression (Soulé & Mills, 1998) and initial population sizes (Fagan & Holmes, 2006) both influence the probability of population extinction. Hundreds of empirical studies indicate that population genetic structure is magnified in fragmented habitats due to restricted gene flow (Bohonak, 1999; Keyghobadi, 2007), but many of these studies do not test explicit population genetic hypotheses (Emel & Storfer, 2012). Additionally, the interacting roles of population density, fragment area, habitat quality, and spatial configuration in driving loss of genetic variation vary widely across taxa or ecosystems (Gibbs, 2001; Fahrig, 2003).

Severe fragmentation is a typical fate of native remnant habitats in cities (Shochat et al., 2006), and urban wildlife with limited dispersal ability are predicted to exhibit genetic differentiation between urban habitat patches (often city parks or similar semi-natural green infrastructure). A growing body of “urban conservation genetics” (Noël & Lapointe, 2010) literature has documented genetic structure between populations of multiple city-dwelling taxa, including mammals (Wandeler et al., 2003; Munshi-South & Kharchenko, 2010; Chiappero et al., 2011), amphibians (Hitchings & Beebee, 1997; Noel et al., 2007; Munshi-South, Zak & Pehek, 2013), reptiles (Delaney, Riley & Fisher, 2010), birds (Bjorklund, Ruiz & Senar, 2010; Vangestel et al., 2011; Unfried, Hauser & Marzluff, 2013), and insects (Watts et al., 2004; Jha & Kremen, 2013). These studies reported either stable or reduced genetic variability in urban vs. non-urban habitats, but few examined associations between patch attributes and population genetic indicators. Larger urban parks harbor increasingly greater numbers of species (Goddard, Dougill & Benton, 2010; Strohbach, Lerman & Warren, 2013), and may also protect individual populations against genetic bottlenecks, inbreeding, and loss of genetic variation. Understanding the relationship between park size and genetic variation will aid efforts to manage networks of small urban patches (Millard, 2008; Vergnes, Viol & Clergeau, 2012).

In this study, we examine whether isolation in New York City (NYC) parks results in genetic bottlenecks in white-footed mice (Peromyscus leucopus). We then use nonlinear regression and information-theoretic model selection to test the hypotheses that park size and time since isolation are associated with genetic variability. Our previous research on this system found substantial genetic structure among urban white-footed mice, and indicators of genetic variation at neutral microsatellites were moderately high but not uniform across parks (Munshi-South & Kharchenko, 2010). In contrast, P. leucopus in fragmented woodlots surrounded by agricultural matrix exhibit only weak genetic structure and high genetic variability (Mossman & Waser, 2001). Emigration rates from small patches may be higher than from large patches in these P. leucopus metapopulations (Anderson & Meikle, 2010), presumably because the smallest patches contain the highest population densities (Krohne & Hoch, 1999). Our previous estimates of both recent and historical migration between parks were very low between most pairs of parks in NYC (Munshi-South, 2012). Thus, the probability of bottlenecks and levels of genetic variation in urban white-footed mice should be influenced more by park size and how long the sites have been isolated than by migration rates. The short timeframe of urbanization in NYC also indicates that mutations will be a weak contributor to contemporary genetic diversity.

Natal dispersal in most mammals (Greenwood, 1980; Dobson, 1982), including P. leucopus (Wolff, Lundy & Baccus, 1988), is male-biased. This pattern may result in lower average relatedness and weaker genetic structure between members of the dispersing vs. philopatric sex (Mossman & Waser, 1999; Munshi-South, 2008). In urban populations, we predict that a male bias in dispersal will be weak to nonexistent due to an inability for either sex to successfully disperse out of isolated urban patches. We use neutral microsatellite data and four different statistical tests to assess this prediction. Given that sex-biased dispersal may create disparities between population genetic patterns inferred from bi- vs. uni-parentally inherited markers, we also sequenced a 324 bp segment of the mitochondrial D-loop. We use these sequence data for independent inferences of population demography and genetic variation of urban P. leucopus. Specifically, we used mismatch distribution analyses to statistically assess the evidence for a population expansion after a bottleneck or selective sweep in NYC parks, and estimate the number of generations since any such events.

Urban biodiversity is increasingly recognized as worthy of conservation attention (Elmqvist et al., 2013), but the population genetics of wildlife in cities has received relatively little attention (Magle et al., 2012). This study is one of the first to examine population bottlenecks, genetic variation, and sex-biased dispersal of wildlife in relation to the characteristics of urban parks.

Methods

Sampling and microsatellite data collection

To examine associations between urban park size and genetic variation, we trapped and sampled genetic material from 294 white-footed mice from 14 urban parks in NYC from 2008 to 2009. These study sites encompass nearly all of the large forested areas known to harbor P. leucopus in the NYC boroughs of the Bronx, Manhattan, and Queens. Brooklyn and Staten Island were excluded from the study a priori due to logistical constraints. The trapping sites within each park were usually located in “Forever Wild” nature preserves that are protected to maintain urban biodiversity and ecosystem services. These “Forever Wild” sites are situated within a broader park matrix of mowed lawns, playgrounds, athletic fields and other managed landscapes. Most trapping sites consisted of an invasive understory and an Appalachian oak-hickory or successional northern hardwoods canopy as defined by Edinger et al. (2002), with successional shrublands, oldfields, and salt marsh edges at three Queens sites (Fort Tilden, Willow Lake, and Jamaica Bay, respectively; Table 1).

Table 1 Characteristics of study sites and results of bottleneck tests.

Total area of site, area of potential white-footed mouse habitat, percent habitat, and years since park founding (a proxy for isolation time) for 14 NYC parks analyzed in this study. Site abbreviations follow Fig. 1. Final column represents the P-value calculated from 10,000 randomizations of the bottleneck test. Significant values appear in bold.

Site	Borough	Total area
(ha)	Habitat area
(ha)	Percent
habitat	Years since
founding	Bottleneck
P-value	
Hunters Island (HI)	Bronx	247.23	103.47	0.42	121	0.71	
NY Botanical Garden (NYBG)	Bronx	98.23	37.44	0.38	114	0.838	
S. Pelham Bay (SPel)	Bronx	126.24	64.06	0.51	121	0.567	
Van Cortlandt Park (VC)	Bronx	433.15	226.83	0.52	121	0.29	
Central Park (CP)	Manhattan	344.05	45.23	0.13	136	0.011	
Inwood Hill Park (In)	Manhattan	79.21	52.53	0.66	93	0.935	
Alley Pond Park (AP)	Queens	219.66	164.26	0.75	82	0.033	
Cunningham Park (CH)	Queens	188.31	123.50	0.66	71	0.517	
Willow Lake (FM)	Queens	42.09	25.84	0.61	75	0.009	
Forest Park (FP)	Queens	230.68	129.84	0.56	114	0.433	
Fort Tilden (FT)	Queens	248.96	66.71	0.27	92	0.416	
Jamaica Bay (JB)	Queens	263.38	263.38	1.00	71	0.071	
Kissena Park (KP)	Queens	61.44	17.68	0.29	103	0.959	
Ridgewood Reservoir (RWR)	Queens	50.58	28.40	0.56	103	0.695	

Mice were trapped over 2–3 nights at each site using Sherman Live Traps (9″ × 9″ × 3″) baited with birdseed. For genetic analysis, we snipped the terminal 1 cm or less of each mouse’s tail before releasing them alive at the site of capture. Tail snips were stored in 80–95% ethanol until DNA extraction. Next, we genotyped all mice at 18 unlinked microsatellite loci, and calculated for each population across all loci the mean number of alleles, effective number of alleles (i.e., the estimated number of equally frequent alleles in an ideal population), number of private alleles, and observed heterozygosity in GenAlex 6.2 (Peakall & Smouse, 2006). All animal handling procedures were approved by the CUNY Brooklyn College Institutional Animal Care and Use Committee (Protocol No. 229). Permission to collect genetic samples from wild white-footed mice was granted by the New York State Department of Environmental Conservation (License to Collect or Possess Wildlife Nos. 1262 and 1603), Gateway National Recreation Area, the NYC Department of Parks and Recreation, and the Central Park Conservancy. Full descriptions of study sites, microsatellite loci, genotyping protocol, and calculation of basic population genetic statistics are available in Munshi-South & Kharchenko (2010). The microsatellite genotypes are also available on the Dryad Digital Repository (DOI 10.5061/dryad.1893).

Analysis of historical demography and sex-biased dispersal using microsatellite loci

We tested for genetic bottlenecks in each park using the program BOTTLENECK 1.2 (Piry, Luikart & Cornuet, 1999) and the authors’ recommended settings for microsatellites (two-phase mutations; 95% single-step and 5% multi-step mutations). We report results of a one-tailed Wilcoxon’s signed rank test based on 10,000 randomizations to examine the hypothesis of significant heterozygosity excess in bottlenecked populations (Cornuet & Luikart, 1996).

To test for sex-biased dispersal between NYC parks in urban white-footed mice, we used the “biased dispersal” module in FSTAT 2.9.3 (Goudet, Perrin & Waser, 2002) to compare multiple indices between males and females: the mean and variance of a corrected assignment index (AIc), average relatedness, and FST calculated separately for males and females. The assignment index calculates the probability that an individual’s genotype occurred by chance in a population, and thus individuals of the dispersing sex should exhibit lower mean AIc values (Paetkau et al., 1995; Favre et al., 1997). We used one-sided P values calculated using 10,000 randomizations of the data to test the predictions of lower mean AIc, greater variance in AIc, lower average relatedness, and lower FST for males than for females. These indices were calculated for the entire dataset, as well as for two subsets of populations: Bronx and Queens. Previous work indicated that migration rates are nonzero between at least some populations in each subset (Munshi-South, 2012), but results were not different from the total dataset.

Modeling of park size vs. genetic diversity

We modeled five basic measures of genetic variation against area and time since isolation of each of the 14 NYC parks to examine the effect of park attributes on genetic diversity. Genetic measures included the mean number of alleles (NA), effective number of alleles (NE), number of private alleles (NP), and observed heterozygosity (HO) reported in Munshi-South & Kharchenko (2010), as well as Θ (4NEμ where μ = mutation rate) estimated using MIGRATE-n (Beerli, 2006) and reported in Munshi-South (2012). These genetic measures were modeled with three geographic covariates: total park area (TA), natural habitat area defined as secondary or primary forest cover (HA), and the proportion of habitat area out of the total area (PH). Geographic layers of park boundaries were obtained from the New York City Department of Parks and Recreation. Habitat delineations were digitized by hand in ArcGIS 10.1 using aerial photographs and our own knowledge of each park’s layout.

We also modeled genetic variability with the number of years since each park became isolated to examine the hypothesis that longer periods of isolation result in lower genetic variability. We used two different proxies for time since isolation: (1) the years since each park was officially founded, and (2) years since major infrastructure projects (primarily multi-lane parkways and expressways) were erected around park perimeters. Our rationale for the former date was that many parks in NYC were likely the last large green spaces in the general area at the time of their establishment, and thus their founding dates should reflect the order in which they became isolated. We used a time since isolation based on infrastructure projects because many NYC parks are at least partially circled by major roadways that likely present major obstacles to wildlife. Networks of parkways were constructed during the Robert Moses era of park management in NYC (Caro, 1974), and these parkways may have been the most important factor in loss of connectivity between populations. Information that would facilitate more precise inference of the time since isolation is generally not available. Aerial photos of NYC from 1924 and 1951 (Available at http://maps.nyc.gov/doitt/nycitymap/) indicate that most parks were surrounded by urban development by 1924, and thus isolated concurrent with or before their founding. Many maps exist from throughout NYC’s history, but unfortunately these maps generally do not contain information on the quality or extent of vegetation (Benson, 2013). Generating habitat cover through more complex predictive approaches for even one snapshot of time in NYC is a monumental effort outside the scope of this study (Sanderson, 2009).

Genetic (dependent) variables were examined with eighteen candidate models, each of which consisted of various combinations of the three geographic covariates and time since founding (F): an intercept-only model, the four univariate models (TA, HA, PH, and F), the eight combinations of two covariates,the four combinations of three covariates, and a global model with all four covariates (TA + HA + PH + F). Intercept-only models were included in the candidate model sets of each genetic variable to serve as a baseline for detecting a covariate effect: if a model performed substantially better than the intercept-only model, we interpreted this result as evidence for an effect of that model’s covariates upon the respective diversity index. We calculated maximum likelihood estimates of model parameters for each model, and then models were ranked using Akaike’s Information Criterion (AICc) corrected for finite sample sizes (Burnham & Anderson, 2002). In brief, this approach compares a set of models, each representing an a priori hypothesis, to determine which model is closer to a hypothetical model that encompasses all of reality, i.e., one that perfectly models the dependent variable in all instances. The advantages and general differences of an information-theoretic approach versus traditional hypothesis testing were discussed by Anderson, Burnham & Thompson (2000).

Each of the six genetic diversity indices was modeled with regression techniques appropriate to the distribution of that index, based on the overall sample frequency distribution, e.g., right-skewed variables used gamma regression. If we were unsure of the proper regression method between a choice of two, the method that minimized the deviance of the global model was used. Traditionally, a transformation (e.g., log, square-root) is applied to non-normal data to facilitate the use of regression; however, this approach may be inferior to using regression techniques that directly match the distributions of the variables in question (Gea-Izquierdo & Cañellas, 2009). All modeling was performed in R 2.15 (R Development Core Team, 2012). Gamma and GLM regression were specifically performed using the MASS package (Venables & Ripley, 2002), and analyses with a Tweedie distribution using the “tweedie” package (Dunn, 2013).

Mitochondrial DNA sequencing and analyses

We sequenced a 324 bp region of the mitochondrial D-loop for a subset of 110 individuals from above to examine historical demography and genetic variation using a maternally inherited marker. We designed D-loop PCR primers from a consensus sequence created using all P. leucopus D-loop sequences available on GenBank in September 2009 (accession numbers available from authors upon request). We created the consensus sequence from a ClustalW alignment conducted in BioEdit 7 (Hall, 1999), and chose the primers using the Primer3 web interface (Rozen & Skaletsky, 1999). We conducted PCR in 25 µl volumes using Illustra PuReTaq Ready-to-Go PCR beads (G.E. Life Sciences, Piscataway, NJ) with one µl forward primer (Pleucopus_DloopFor 5′-ACCATCCTCCGTGAAATCAG-3′), one µl reverse primer (Pleucopus_DloopRev 5′-AAAAAGCATATGAGGGGAGTG-3′), and one µl of template DNA with concentrations of 25–50 ng/µl. We performed PCR on a thermocycler for 30 cycles of 95 °C for 30 s, 55 °C for 30 s, and 72 °C for 1 min, and then cleaned PCR products using Qiaquick PCR purification kits (Qiagen, Valencia, CA). We then sequenced both forward and reverse strands using the standard GenomeLab DTCS quick start protocol on a Beckman Coulter CEQ 8000 sequencer (Beckman Coulter, Brea, CA). Finally, we edited and aligned the sequences using Sequencer 4.8 (Gene Codes, Ann Arbor, MI) and BioEdit 7. All unique, unaligned D-loop haplotypes have been deposited on GenBank (Accession: KF986735–KF986771), and a Nexus haplotype file used for the analyses below is available on the Figshare digital repository (DOI 10.6084/m9.figshare.881830).

We calculated summary statistics for all D-loop sequences and subsets from Bronx, Manhattan, and Queens using DnaSP 5.1 (Rozas et al., 2003). Statistics we used to describe D-loop variation included the number of polymorphic sites, nucleotide diversity, number of haplotypes, haplotype diversity, and the average number of nucleotide differences. To examine deviations from neutrality and population size changes, we also calculated Tajima’s D and Fu’s Fs, and assessed their significance using 10,000 coalescent simulations. We also calculated mismatch distributions (i.e., the observed pairwise nucleotide site differences) under a model of population expansion to examine demographic changes, and assessed significance of the observed distributions using 10,000 coalescent simulations of the raggedness statistic, r (Rogers & Harpending, 1992), and the R2 statistic (Ramos-Onsins & Rozas, 2002).

Results & Discussion

Analysis of historical demography and sex-biased dispersal using nuclear loci

Tests for genetic bottlenecks did not detect significant heterozygosity excess in most NYC parks (Table 1), indicating that bottlenecks have not been a general phenomenon in these populations. Three populations tested positive for recent bottlenecks, but there was no general trend towards bottlenecks in large or small parks (Table 1). The estimated habitat area (HA) was among the lowest of the 14 parks for two of the populations exhibiting bottlenecks (Central Park and Willow Lake), but other small habitat patches were not positive for bottlenecks. These results suggest that very small urban forest fragments (e.g., <50 ha) may support sufficiently large populations of native small mammals to prevent severe genetic drift from population crashes. The relatively high population densities of P. leucopus that have been recorded in small patches may explain this resiliency (Krohne & Hoch, 1999). However, the lack of evidence for bottlenecks is still surprising given that substantial genetic drift has occurred in these populations over the past century (Munshi-South & Kharchenko, 2010), and drift efficiently reduces allelic diversity in isolated populations (Allendorf, 1986). This type of analysis should be interpreted cautiously because the results may be influenced by deviations from the underlying assumptions about microsatellite mutation rates. Many of our microsatellite markers were first identified in the closely-related P. maniculatus, and thus in P. leucopus may not strictly adhere to a stepwise mutation model with a low frequency of multi-step mutations. These microsatellites generally adhered to other expectations (e.g., Hardy–Weinberg equilibrium) and performed well in a number of other analyses, and thus we feel it is unlikely that we are failing to detect true bottlenecks in these populations. Changes to the assumed frequencies of single- vs. multi-step mutations in our Bottleneck runs did not significantly alter the results.

We also found little evidence of sex-biased dispersal in urban white-footed mice, either across all NYC populations or clustered sites in Bronx or Queens (Table 2). Only one of four statistics varied in the predicted direction for male-biased dispersal (mean AIc), but this sex difference was not robustly supported. Males in our sample had less likely genotypes than females given the overall genetic characteristics of our sample, as has been argued previously to support male-biased dispersal in this species (Mossman & Waser, 1999). Our previous findings of very low migration rates between urban populations (Munshi-South, 2012) coupled with these dispersal results suggest that neither males nor females migrate between urban patches at anywhere near the high rates reported for less severe fragmentation scenarios (Anderson & Meikle, 2010). However, males may still disperse more often or farther away from their natal sites within patches than females disperse. Our study design and sample sizes for each site did not allow us to test within-patch dispersal patterns.

Table 2 Tests for male-biased dispersal in urban white-footed mice.

Results of sex-biased dispersal analysis for white-footed mice across all 14 NYC parks, a subset of six parks in Bronx, and a subset of parks in Queens.

	N	Mean AIc	Variance AIc	Relatedness	FST	
All NYC Parks	301	–	–	0.14	0.08	
Females	165	0.43	31.8	0.15	0.09	
Males	136	−0.52	32.2	0.14	0.08	
Bronx	104	–	–	0.16	0.09	
Females	52	0.47	35.7	0.15	0.09	
Males	52	−0.47	36.8	0.15	0.08	
Queens	157	–	–	0.14	0.08	
Females	90	0.40	31.8	0.15	0.09	
Males	67	−0.53	26.3	0.13	0.08	

Few studies have examined bottlenecks or sex-biased dispersal in species isolated in urban forest fragments. Our results for white-footed mice suggest that other small vertebrates with limited dispersal ability (especially non-volant species) can avoid genetic bottlenecks if they maintain high population densities in small urban parks. Common forest dwellers in eastern North America that are known to thrive in urban parks include red-backed salamanders, Plethodon cinereus (Noël & Lapointe, 2010), and northern short-tailed shrews, Blarina brevicauda (Brack Jr, 2006). The former species responds similarly to urban forest fragmentation as P. leucopus in terms of rapid genetic differentiation between fragments but little apparent loss of genetic diversity (Gibbs, 1998; Noel et al., 2007). We predict that B. brevicauda will exhibit similar patterns. In contrast, the northern dusky salamander, Desmognathus fuscus, loses genetic variation in isolated urban stream/seepage habitats (Munshi-South, Zak & Pehek, 2013), but this pattern may not hold for species such as the northern two-lined salamander, Eurycea bislineata, that maintain higher population densities and occupy a greater diversity of streams in urbanized watersheds (Pehek, 2007).

Modeling of park characteristics vs. genetic diversity

The values for NA, NE, and NP were right skewed continuous and thus were modeled using gamma regression (Gea-Izquierdo & Cañellas, 2009), while HO was best modeled with a standard generalized linear model (GLM). Θ was highly right-skewed and thus was best modeled using a Tweedie distribution with an inverse Gaussian dispersion parameter (i.e., p = 3). The inverse Gaussian parameterization of the generalized Tweedie distribution is useful in modeling variables that are right-skewed and continuous (Jorgensen, 1987; Dunn & Smyth, 2001). In all model sets the global model deviance/degrees of freedom were less than or equal to 1.0, indicating adequate model fit (i.e., no overdispersion). For all diversity measures except Θ the intercept-only models had the most parsimonious fit (Table 3), indicating no discernible pattern between the geographic covariates/time since park founding and the genetic diversity measures (Fig. 1). Θ increased as the percent of habitat area in each park increased (Table 3; Fig. 1S). The model analyzing Θ with percent habitat and time since founding was also highly ranked, but due to the effect of percent habitat rather than time since founding. Two geographic covariate pairs—total area (TA) and habitat area (HA), and HA and percent habitat (PH)—were correlated (r = 0.68 and 0.63, respectively); however, models with these combinations were universally poor.

Figure 1 Scatterplots of genetic variation vs. characteristics of NYC parks.

Scatterplots of observed heterozygosity (A–D), number of alleles (E–H), number of effective alleles (I–L), number of private alleles (M–P), and Θ (4NEμ; Q–T) on the y-axis vs. (from left to right) total park area (ha), habitat area (ha), percent habitat, and years since founding on the x-axis. Each of 14 NYC parks is labeled within each scatterplot with an abbreviation following Table 1.

Table 3 Model selection for park characteristics vs. genetic diversity.

Results of model selection for park characteristics vs. genetic diversity indices.

Model	LogLike	k	ΔAICc	Model	LogLike	k	ΔAICc	
Number of alleles (NA)	Effective number of alleles (NE)	
Intercept	−25.91	2	0.00	Intercept	−15.20	2	0.00	
TA	−25.54	3	2.58	F	−15.13	3	3.17	
HA	−25.61	3	2.71	HA	−15.15	3	3.21	
PH	−25.85	3	3.19	HA ∗ F	−10.60	5	3.21	
F	−25.87	3	3.22	TA	−15.15	3	3.22	
HA ∗ F	−22.49	5	5.56	PH	−15.19	3	3.29	
TA + F	−25.35	4	6.23	PH + F	−14.98	4	6.91	
TA + PH	−25.45	4	6.43	HA + F	−15.04	4	7.03	
TA + HA	−25.51	4	6.55	TA + F	−15.11	4	7.18	
HA + PH	−25.60	4	6.73	TA + HA	−15.14	4	7.24	
HA + F	−25.60	4	6.74	HA + PH	−15.14	4	7.24	
PH + F	−25.85	4	7.23	TA + PH	−15.14	4	7.25	
TA ∗ F	−24.43	5	9.45	TA ∗ F	−13.60	5	9.22	
TA + HA + F	−25.33	5	11.25	HA + PH + F	−14.97	5	11.96	
TA + PH + F	−25.35	5	11.28	TA + PH + F	−14.97	5	11.96	
TA + HA + PH	−25.36	5	11.31	TA + HA + F	−14.99	5	11.99	
HA + PH + F	−25.53	5	11.65	TA + HA + PH	−15.14	5	12.30	
Global	−25.26	6	17.60	Global	−14.97	6	18.46	
Number of private alleles (NP)	Observed heterozygosity (HO)	
Intercept	−33.40	2	0.00	Intercept	25.75	2	0.00	
HA	−33.29	3	3.07	PH	26.13	3	2.56	
PH	−33.31	3	3.12	F	26.11	3	2.60	
TA	−33.40	3	3.31	HA	26.03	3	2.75	
F	−33.40	3	3.31	TA	25.75	3	3.31	
TA + HA	−33.21	4	6.98	TA + HA	26.49	4	5.87	
PH + F	−33.22	4	6.99	HA + F	26.38	4	6.10	
HA + PH	−33.27	4	7.09	HA + PH	26.20	4	6.45	
HA + F	−33.28	4	7.11	PH + F	26.20	4	6.46	
TA + PH	−33.30	4	7.16	TA + F	26.17	4	6.52	
TA + F	−33.40	4	7.35	TA + PH	26.13	4	6.59	
HA ∗ F	−32.59	5	10.78	HA ∗ F	27.74	5	8.44	
TA ∗ F	−32.88	5	11.35	TA ∗ F	26.70	5	10.51	
TA + HA + F	−33.07	5	11.74	TA + HA + PH	26.65	5	10.61	
TA + HA + PH	−33.11	5	11.83	TA + HA + F	26.53	5	10.85	
HA + PH + F	−33.21	5	12.03	HA + PH + F	26.38	5	11.16	
TA + PH + F	−33.22	5	12.04	TA + PH + F	26.23	5	11.44	
Global	−33.03	6	18.15	Global	26.80	6	16.81	
Θ					
HA ∗ F	–*	–	–					
Global	–*	–	–					
PH	−39.69	3	0.00					
PH + F	−38.99	4	2.63					
Intercept	−43.10	2	3.51					
HA + PH	−39.44	4	3.54					
TA + PH	−39.50	4	3.66					
HA	−42.27	3	5.15					
TA + HA	−40.32	4	5.30					
F	−42.57	3	5.76					
HA + F	−40.68	4	6.01					
TA + F	−40.82	4	6.30					
TA	−43.09	3	6.79					
TA + PH + F	−38.98	5	7.67					
HA + PH + F	−38.98	5	7.68					
TA + HA + PH	−39.44	5	8.59					
TA + HA + F	−39.64	5	8.99					
Notes.

TA total area of park

HA undeveloped habitat area of park

PH proportion of park habitat area to total park area

F years since founding of park

* Denotes model that did not converge.

While there was considerable variation in measures of allelic diversity among NYC parks, there was no clear relationship between park size/time since founding and genetic variation. Neither estimate of the time since founding was successful at explaining genetic variation, and we only report the results from the time since founding (Table 3, Fig. 1). Observed heterozygosity did not differ very much between most populations. This latter result suggests that even the smallest and most isolated habitat patches maintain population densities that are adequate to preserve heterozygosity. Additionally, the parks we sampled in NYC may all fall below a size threshold beyond which white-footed mice maintain high population densities and genetic variation, sometimes referred to as a “synurbic” threshold (Francis & Chadwick, 2012). Larger urban parks support higher population densities of gray squirrels (Sciurus carolinensis), although the relative proportions of tree and building cover also influence these densities (Parker & Nilon, 2012). Most of the parks analyzed in this study contained similar types of mouse habitat: an invasive vegetative understory with an oak-hickory or successional northern hardwoods canopy (Munshi-South & Kharchenko, 2010). However, it is possible that unmeasured ecological variability between NYC parks, such as differences in habitat quality, food availability, or predator abundance (Levi et al., 2012), would better explain genetic variation than park size or the time since founding. Genetic variation in P. leucopus may alternatively respond to ecological variables in non-generalizable ways. For example, competition between squirrels, chipmunks (Tamias striatus), and white-footed mice in northeastern forest fragments is weak to nonexistent, except in certain sites with idiosyncratic characteristics (Brunner et al., 2013).

NYC parks with the largest habitat areas may be instructive about the influence of highly site-specific characteristics on genetic diversity. Jamaica Bay (JB) had one of the largest habitat areas but the lowest measures of genetic diversity (Fig. 1). This site differs from the others in that the habitat is composed of salt marsh and sandy scrub in addition to forests, and thus may be lower-quality habitat than the typical urban forest. We ran the modeling procedure without Jamaica Bay, but the overall results did not change. In contrast to Jamaica Bay, Van Cortlandt (VC) is one of the largest parks in NYC and also exhibited the highest allelic diversity. Besides large size, Van Cortland contains a diversity of forest and meadow habitats, and roads may promote weak genetic differentiation between mice in the park (Munshi-South & Kharchenko, 2010). The remaining parks may have not sufficiently differed in size and genetic diversity to identify a general trend, although the Van Cortland results suggest that very large urban parks may harbor the greatest genetic variation if they are diverse in vegetation and structure.

Park isolation time was generally not successful at explaining genetic variation. Similar to park size, there may not be enough variation in the time since isolation of NYC parks to observe a general trend in genetic variation. Alternatively, our use of the years since a park was founded or the years since construction of major infrastructure as proxies for how long parks have been ecologically isolated may not have been accurate. Given that much of NYC was heavily urbanized concurrent with the founding of these parks (particularly outside Manhattan), we feel that our choices were justified. However, future historical reconstructions of the NYC landscape (Sanderson, 2009) may fruitfully revisit this question. For the specific question of ecological and genetic isolation of wildlife, some parks may also have a complicated history of human transformation that will be difficult to account for in studies such as ours. For example, both Central Park in Manhattan and Flushing Meadows-Willow Lake in Queens are largely human-made habitats constructed after periods of heavy human disturbance (farms and villages in Central Park, and a massive ash dump in Flushing Meadows). Thus, it is not clear whether white-footed mouse populations have always been present at these sites, or recolonized after some period of absence. Relatively high levels of genetic variation suggest the former scenario, but definitive historical trapping records are not available.

Mitochondrial DNA, demographic changes, and selection

We identified 37 haplotypes among 324 bp mitochondrial D-loop sequences obtained for 110 individuals (Table 4). Haplotype diversity was very high, but the average number of nucleotide differences between haplotypes was low to moderate. Analysis of haplotypes by landmass (Bronx, Manhattan & Queens) revealed similar patterns, although Queens exhibited lower diversity and average number of differences than the other two landmasses or the total sample, despite a larger number of haplotypes and polymorphic sites (Table 4).

Table 4 Statistical analysis of 324 bp of the mtDNA D-loop from 110 white-footed mouse individuals.

Population	N a	P b	π c	H d	Hd e	K f	D g	Fs h	r i	R2 j	τ k	
Bronx	26	13	0.0089	12	0.91	2.70	−0.71	−4.38*	0.04	0.099	2.70	
Manhattan	22	10	0.0113	7	0.82	3.61	1.08	0.81	0.13	0.177	2.93	
Queens	56	22	0.0051	17	0.74	1.62	−2.1**	−11.5**	0.029*	0.041**	0.88	
All	110	30	0.0084	37	0.91	2.51	−1.73*	−34.29**	0.058	0.04*	2.09	
Notes.

Significant values are presented in bold text at *P < 0.05 or **P < 0.01 based on 10,000 coalescent simulations in DNASP.

a Number of individuals haplotyped.

b Number of polymorphic sites in D-loop sequence.

c Nucleotide diversity.

d Number of D-loop haplotypes.

e Haplotype diversity.

f Average number of pairwise nucleotide differences.

g Tajima’s D.

h Fu’s Fs.

i Raggedness statistic for mismatch distribution.

j Ramos-Onsins & Rozas R2 statistic for mismatch distribution.

k τ (2μt) calculated for mismatch distribution.

Significant Tajima’s D and Fu’s Fs values for the total NYC sample and Queens indicate that urban P. leucopus underwent a recent population expansion after a bottleneck or selective sweep. Alternatively, the D-loop may have experienced genetic hitchhiking (i.e., genetic draft) due to negative selection on linked mitochondrial genes. The observed mismatch distributions for NYC and Queens closely fit the expected unimodal distribution for a recent population expansion (Fig. 2). This fit was statistically supported by the raggedness and R2 statistics (Table 4). Statistics indicating a demographic expansion were generally not significant for the Bronx and Manhattan subsamples, but this discrepancy may be due to a much smaller sample size for these areas of the city. Environmental or demographic stochasticity can exert considerable influence on mismatch distributions, and such an effect would be enhanced for small sample sizes.

Figure 2 Mitochondrial mismatch distribution analyses for white-footed mice in NYC that show the influence of a recent population expansion after a bottleneck of selective sweep.

Mismatch distributions for 324 bp segment of the mtDNA D-loop for Queens (N = 56; top graph) and all NYC samples (N = 110; bottom graph). The solid line indicates the observed distribution, and the dotted line indicates the expected distribution for a demographic expansion.

We estimated the time (in generations) since the bottleneck or selective sweep reflected in the mismatch distributions (Fig. 2) using the τ parameter (2 µt; Table 4) and four pedigree-derived estimates of the mitochondrial D-loop mutation rate (3.52 × 10−5, 1.92 × 10−5, 1.28 × 10−5, and 4.19 × 10−6/site/generation) in humans (Santos et al., 2005). To our knowledge, no similar D-loop mutation rate estimates have been published for rodents. The four human mutation rates were multiplied by 324 bp, and then used to calculate t = τ/2μ for Queens (38.5, 70.6, 105.9, and 323.9 generations, respectively) and the entire NYC sample (91.5, 167.8, 251.7, and 769.8 generations). These results suggest that demographic expansion after a bottleneck or selective sweep was concurrent with urbanization of NYC (i.e., in the last few hundred years), assuming a conservative generation time in urban P. leucopus between 0.5 and 1.0 years. The expansion also occurred more recently in Queens than NYC overall, potentially because Queens was not heavily urbanized until after the construction of bridges, tunnels, and commuter rail connecting Queens to Manhattan in the early twentieth century. However, these estimated times should be interpreted cautiously given that we used D-loop mutation rates estimated from human pedigrees. These human estimates were calculated over sequences that were a few hundred bp longer than those analyzed in this study. If mutation rates are heterogeneous along the D-loop, then these human estimates may over- or under-estimate the mutation rate for P. leucopus. Additionally, mitochondrial mutation rates vary widely across mammals with variation in body size, temperature, metabolic rates (Gillooly et al., 2005), age at female sexual maturity, and lifespan (Nabholz, Glémin & Galtier, 2008). All of these factors predict that P. leucopus will have a higher mitochondrial mutation rate than humans, and thus our reported times since a bottleneck or selective sweep are likely all underestimated. If mutation rates are substantially higher in P. leucopus, then the timing estimates could indicate that the demographic event or sweep occurred considerably more recently than urbanization/isolation of NYC Parks.

In the absence of independent evidence, it is difficult to distinguish between bottlenecks and selection as explanations for the mismatch distributions observed in this study. Only a few NYC populations exhibited evidence of recent bottlenecks at nuclear microsatellite loci, thus undermining the bottleneck argument for the mtDNA data. Alternatively, the mtDNA results may reflect a bottleneck that occurred further back in time than could be detected by the microsatellite data, or the mtDNA reflects a bottleneck specific to this matrilineal marker. These alternatives are unlikely given that the D-loop is also a hypervariable marker appropriate for detecting recent demographic events, and the events creating the mismatch distribution were estimated to occur in the last few dozens to hundreds of generations. A matrilineal bottleneck signature is also unlikely given that males and females did not differ at multiple population genetic parameters estimated using nuclear data (Table 2).

The mismatch distributions may be better explained by negative selection against unfavorable mtDNA haplotypes once these populations became isolated in urban habitat patches with novel selection pressures. In this scenario, the D-loop would have hitchhiked along with the surviving haplotypes containing favored alleles in mtDNA protein-coding regions. The mitochondrial genome is now widely acknowledged to have experienced selective sweeps in many if not most taxa, and thus some authors have called into question its utility in demographic estimation (Bazin, Glémin & Galtier, 2006; Nabholz et al., 2008; Balloux, 2010). Theoretical arguments and some empirical results show that factors such as changes in population density (Lankau & Strauss, 2011) and increased temperature (Franks, Weber & Aitken, 2014; Schilthuizen & Kellermann, 2014) are likely to produce evolutionary responses in human-altered environments (Sih, Ferrari & Harris, 2011; Mueller et al., 2013). These factors or others related to metabolism (such as a dietary shift in urban habitats) may have driven mitochondrial selective sweeps in NYC’s white-footed mice.

Mitochondrial DNA sequencing of contemporary and museum specimens of P. leucopus from the Chicago area indicated that mtDNA haplotypes changed rapidly over a timeframe corresponding to human development of natural areas (Pergams, Barnes & Nyberg, 2003). Mismatch distributions of D-loop haplotypes from these populations (Pergams & Lacy, 2008) closely resemble those presented here for NYC mice, indicating that mitochondrial selection during urbanization may have been a general phenomenon throughout the range of P. leucopus. Pergams & Lacy (2008) argued that these patterns in the Chicago area were due to replacement of the original residents by migrants with a selective advantage; this scenario seems less likely than selection on standing mtDNA variation in NYC because of the high isolation of NYC populations (Munshi-South, 2012). The case for selection on mtDNA is further bolstered by our recent finding of several genes exhibiting elevated signatures of selection in urban vs. rural P. leucopus populations in the NYC area (Harris et al., 2013). Two of these candidate nuclear genes encode mitochondrial proteins (39S ribosomal protein L51 and Camello-like protein 1), suggesting that mitonuclear pathways may be active targets of selection in urban populations (Dowling, Friberg & Lindell, 2008). Full mitochondrial genome sequences and large-scale mRNA-Seq datasets for urban and rural populations of white-footed mice can be used in the future to examine potential mitonuclear associations.

Conclusions

We report here that isolation in urban parks does not necessarily result in genetic bottlenecks or substantial loss of genetic variation in urban wildlife populations. White-footed mouse populations in larger urban parks, or parks that have been isolated for fewer years, do not generally contain greater genetic variation than smaller or older parks, although we could not address site-specific variability between parks that may exert a greater influence on genetic variation than size alone. These results should be encouraging to conservation biologists working in human-dominated landscapes, as even small networks of green spaces may be sufficient to maintain self-sustaining populations and evolutionary potential of some native species (Goddard, Dougill & Benton, 2010). We also found that isolation in urban parks results in weak to nonexistent sex-biased dispersal in a species known to exhibit male-biased dispersal in less fragmented environments. The breakdown of this dispersal mechanism likely explains the pervasive genetic differentiation among P. leucopus populations in different NYC parks. In contrast to nuclear loci, mitochondrial D-loop haplotypes exhibited a mutational pattern of demographic expansion after a recent bottleneck or selective sweep. Estimates of the timing of this expansion indicate that it occurred concurrent with urbanization of NYC over the last few dozens to hundreds of years. Given the general non-neutrality of mtDNA in many systems and evidence of selection on coding sequences in urban P. leucopus, we argue that the P. leucopus mitochondrial genome experienced recent negative selection against haplotypes not favored in isolated urban parks. In general, rapid adaptive evolution driven by urbanization, global climate change, and other human-caused factors is underappreciated by evolutionary biologists, but many more cases will likely be documented in the near future.

We thank Anna Bernstein and Stephen E. Harris for assistance with the mitochondrial sequencing. William Amos, Seth Magle, Charles Nilon, and two anonymous reviewers provided many constructive comments that greatly improved our manuscript. The New York State Department of Environmental Conservation, the NYC Department of Parks & Recreation, Central Park Conservancy, and New York Botanical Garden graciously provided permission to sample white-footed mice in urban parks.

Additional Information and Declarations

Competing Interests

Author Contributions

Animal Ethics

Field Study Permissions

DNA Deposition

Data Deposition

Christopher Nagy is an employee of the Mianus River Gorge Preserve.

Jason Munshi-South conceived and designed the experiments, performed the experiments, analyzed the data, contributed reagents/materials/analysis tools, wrote the paper, prepared figures and/or tables, reviewed drafts of the paper.

Christopher Nagy analyzed the data, contributed reagents/materials/analysis tools, wrote the paper, prepared figures and/or tables, reviewed drafts of the paper.

The following information was supplied relating to ethical approvals (i.e., approving body and any reference numbers):

All animal handling procedures were approved by the CUNY Brooklyn College Institutional Animal Care and Use Committee (Protocol Number 229). The lead author is currently employed by Fordham University, but completed the field work for this study while employed by CUNY.

The following information was supplied relating to ethical approvals (i.e., approving body and any reference numbers):

Permission to collect genetic samples from wild white-footed mice was granted by the New York State Department of Environmental Conservation (License to Collect or Possess Wildlife Nos. 1262 and 1603), Gateway National Recreation Area, the New York City Department of Parks and Recreation, and the Central Park Conservancy.

The following information was supplied regarding the deposition of DNA sequences:

GenBank KF986735–KF986771.

The following information was supplied regarding the deposition of related data:

NEW: Figshare DOI 10.6084/m9.figshare.881830.

PREVIOUSLY DEPOSITED, REANALYZED DATA: Dryad DOI 10.5061/dryad.1893.

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
