# Peer review of "Urban park characteristics, genetic variation, and historical demography of white-footed mouse (Peromyscus leucopus) populations in New York City"

_PeerJ, doi:10.7717/peerj.310_

## Round 0.1 · original submission · Major Revisions

Your manuscript has now been seen by four reviewers and I have read it myself. My view is that 'medium revision' is probably the correct level, but there is no tick box for this! There seems general agreement that the MS is well-written and interesting, but one reviewer queries the validity of the mtDNA mutation rate estimates and hence timings while a second asks for what I feel would be a useful, new analysis. To these I would add that the authors should be a little more circumspect of the short-comings of both 'Bottleneck', which makes big assumptions about how microsatellites evolve, and mismatch distributions, which include a strong stochastic element. To be acceptable for publication, I would like to see: (i) a fuller exploration of the impact of different mutation rates on the size and variance of the timing estimates; (ii) some attempt to address the issue of park isolation (in my opinion, even crude estimates would be better than none) and (iii) a little more discussion about how the output from 'Bottleneck' is strongly dependent on its input assumptions - if markers are used that were cloned in other species, more interruption mutations are likely and a much bigger deviation from the stepwise mutation model will result. The reviewers together make a number of other useful minor points that should be addressed wherever useful.

Reviewer 1 ·

Basic reporting

No comments

Experimental design

No comments

Validity of the findings

The treatment of the microsatellite data seems unproblematic, but the key inferences drawn from the patterns of variation in the mtDNA are rather speculative. No confidence intervals are presented for any of the mtDNA genetic parameters. This is particularly problematic for the estimation of the timing of the post-bottleneck expansion/selection event the authors claim to detect. Given the relatively short sequence length and low diversity, there is likely to be a large variance in these parameter estimates. Acknowledging that significance tests show that some of these are significantly different from zero, the lack of confidence intervals make it difficult to assess the magnitude of the statistics (and therefore plausible range for the timing of the demographic event). Moreover, 2 of 4 the mutation rates applied to the point estimates of  give time (t) estimates which are likely to be older than urbanisation in NYC (lines 288-292) – on this basis, I am not convinced there is strong evidence to support the conclusion that the event was concurrent (or otherwise) with urbanisation in NYC. Finally applying mutation rates derived in humans to rodents may also be problematic. For instance the mutation rate may vary depending on the fragment size assayed (presumably the human estimates were from full length d-loop sequences while in the current study only 324bp is used), and mutation rates may be higher in rodents given their relatively higher metabolic rates - this might bias time estimates in the current study upwards as the authors may be assuming the clock is running more slowly than it actually is Peromyscus.
The authors are right to point out that it is difficult to distinguish between demographic expansion and a selective sweep with this data (line 296), but they then go on to place the emphasis on selection for the rest of this discussion, which is perhaps a little too speculative given the strength of the data presented.

Additional comments

Overall this is a potentially interesting paper and may give insights into impacts on genetic variation for other species inhabiting fragmented, human dominated landscapes. The study is based on reasonably sized microsatellite and mtDNA dataset, some aspects of which (including summary/descriptive statistics) have been published previously. The data collection and data analysis uses standard methodology and analyses. The manuscript is generally very written, clear and an appropriate length. A few minor points are listed below.

1. Lines 98-109, why chose this method rather than assess migration using Structure or Migrate and compare rates for males and females?
2. Line 113 – define ‘effective number of alleles’
3. Line 274 – “Significantly negative...” – poor phrasing, suggest finding alternative.
4. Line 277 – Typo “...(i.e. genetic draft)...” – correct to “genetic drift”
5. line 330 – More details needed, specify which loci?

·

Basic reporting

I've added a few comments to the draft of the manuscript concerning the introduction and background. The goal of the manuscript is buried in the justification. It might be better to put all of the justification for the study before you state the goals and hypotheses. Then, clearly state the hypotheses:
We tested three hypotheses concerning the the role of.....
1.
2.
3.
I do not think it is necessary to focus on the methods that were used to test or evaluate the hypotheses in the introduction.

This may be a more traditional style or perhaps "old school" style of writing, but I think that it would make your manuscript more accessible to those who do not have a background in conservation genetics.

Experimental design

The broader context of urban parks in NYC is missing from the manuscript. A paragraph describing the parks, the habitats that are important, and their landscape and management setting is important to readers with a focus on management and conservation AND to those who are not familiar with the parks that you describe.

The detailed justification of AIC is not needed and could be replaced by a more detailed justification of the park variables used in your models and of the apriori models.

Validity of the findings

No comments

Additional comments

This is an interesting and manuscript that is relevant to readers with a broad interest in urban ecology and urban wildlife management and conservation. A more clear discussion of how the results are relevant to management and conservation would be useful.

·

Basic reporting

Writing is clear and concise, and nicely draws attention to the importance of the topic. A bit of information from the introduction could perhaps be moved to the methods section (lines 63-70, for example). Literature cited seems thorough.

Experimental design

Sampling and statistical methods are rigorous and appropriate. Research questions are meaningful and important. As I am not a geneticist, I cannot directly comment on the genetic methodology of the paper and I hope that the other reviewer(s) are able to do so. A bit more information on how sites were selected (e.g. randomly, opportunistically, along a gradient, etc.), and how mice were trapped and handled is warranted. I am certain that is covered in the cited work, but a few sentences here would also be useful.

Validity of the findings

I am comfortable with the author’s interpretation of their findings, though again I cannot comment on the specific methodology used for the genetic analysis.

I find it a bit surprising that the intercept-only models dominated the analysis of geographic variables. Even if only by chance I might have expected some of them to have some degree of explanatory power. This result is certainly possible but its unusual nature should perhaps be further explored in a discussion.

In general the discussion is appropriate, speculates in places but does not overreach.

Reviewer 4 ·

Basic reporting

No Comments

Experimental design

The paper tests for genetic isolation based on the size of the fragment (park size), but does not test for genetic isolation based on the length of time the fragment was isolated. Estimated migration rates are not an adequate proxy. Just as we would expect greater genetic isolation with smaller park size, we would also expect greater genetic isolation with greater length of time the park is isolated, and a model incorporating both factors would be of great interest. I would like to see a re-analysis modeling GENETIC DIVERSITY against both PARK SIZE (as was already done) and PARK ISOLATION TIME (the number of years each park was isolated). I understand this will require some actual non-genetic historical research; as well as some estimation and approximation, including possibly arbitrary definitions of how isolation would be defined in this context.

Validity of the findings

It should be emphasized that ..."even small networks of green spaces may be sufficient to maintain self-sustaining populations and evolutionary potential of native species" (Abstract) refers in this case only to mice, very small species indeed. We all know how statements may be taken out of context for other purposes. The authors express this better in lines 210-212.

I also think the validity of the findings is called into question by the authors not having considered park isolation time directly, as I stated above.

Additional comments

In general an excellent paper, but with what I consider to be one moderate-major flaw. It is my hope that you re-analyze using park isolation time as an additional factor to park size, and that I have the pleasure of re-reviewing your fine work.

---

## Round 0.2 · accepted · Accept

In general you have addressed the issues raised well. I still quibble a bit about the reliability of 'Bottleneck' because the program does not account for heterogeneity of mutation rate among alleles. However, this is probably more to do with personal pedantry than something others will worry about!